# Development of Innate-Immune-Cell-Based Immunotherapy for Adult T-Cell Leukemia–Lymphoma

**DOI:** 10.3390/cells13020128

**Published:** 2024-01-10

**Authors:** Maho Nakashima, Yoshimasa Tanaka, Haruki Okamura, Takeharu Kato, Yoshitaka Imaizumi, Kazuhiro Nagai, Yasushi Miyazaki, Hiroyuki Murota

**Affiliations:** 1Department of Dermatology, Graduate School of Biomedical Sciences, Nagasaki University, Nagasaki 852-8501, Japan; 2Center for Medical Innovation, Nagasaki University, Nagasaki 852-8588, Japan; 3Department of Tumor Cell Therapy, Hyogo College of Medicine, Nishinomiya 663-8501, Japan; 4Department of Hematology, Nagasaki University Hospital, Nagasaki 852-8501, Japan; 5Department of Hematology, National Hospital Organization Nagasaki Medical Center, Omura 856-8562, Japan; 6Department of Clinical Laboratory, National Hospital Organization Nagasaki Medical Center, Omura 856-8562, Japan; 7Department of Hematology, Atomic Bomb Disease Institute, Nagasaki University, Nagasaki 852-8523, Japan; 8Leading Medical Research Core Unit, Life Science Innovation, Nagasaki University Graduate School of Biomedical Sciences, Nagasaki 852-8521, Japan

**Keywords:** adult T-cell leukemia–lymphoma, γδ T cell, infusion therapy, interleukin-2, interleukin-18, nitrogen-containing bisphosphonate prodrug, NK cell

## Abstract

γδ T cells and natural killer (NK) cells have attracted much attention as promising effector cell subsets for adoptive transfer for use in the treatment of malignant and infectious diseases, because they exhibit potent cytotoxic activity against a variety of malignant tumors, as well as virus-infected cells, in a major histocompatibility complex (MHC)-unrestricted manner. In addition, γδ T cells and NK cells express a high level of CD16, a receptor required for antibody-dependent cellular cytotoxicity. Adult T-cell leukemia–lymphoma (ATL) is caused by human T-lymphotropic virus type I (HTLV-1) and is characterized by the proliferation of malignant peripheral CD4^+^ T cells. Although several treatments, such as chemotherapy, monoclonal antibodies, and allogeneic hematopoietic stem cell transplantation, are currently available, their efficacy is limited. In order to develop alternative therapeutic modalities, we considered the possibility of infusion therapy harnessing γδ T cells and NK cells expanded using a novel nitrogen-containing bisphosphonate prodrug (PTA) and interleukin (IL)-2/IL-18, and we examined the efficacy of the cell-based therapy for ATL in vitro. Peripheral blood samples were collected from 55 patients with ATL and peripheral blood mononuclear cells (PBMCs) were stimulated with PTA and IL-2/IL-18 for 11 days to expand γδ T cells and NK cells. To expand NK cells alone, CD3^+^ T-cell-depleted PBMCs were cultured with IL-2/IL-18 for 10 days. Subsequently, the expanded cells were examined for cytotoxicity against ATL cell lines in vitro. The proportion of γδ T cells in PBMCs was markedly low in elderly ATL patients. The median expansion rate of the γδ T cells was 1998-fold, and it was 12-fold for the NK cells, indicating that γδ T cells derived from ATL patients were efficiently expanded ex vivo, irrespective of aging and HTLV-1 infection status. Anti-CCR4 antibodies enhanced the cytotoxic activity of the γδ T cells and NK cells against HTLV-1-infected CCR4-expressing CD4^+^ T cells in an antibody concentration-dependent manner. Taken together, the adoptive transfer of γδ T cells and NK cells expanded with PTA/IL-2/IL-18 is a promising alternative therapy for ATL.

## 1. Introduction

Adult T-cell leukemia–lymphoma (ATL) is a mature peripheral CD4^+^ T-cell malignancy caused by infection with human T-lymphotropic virus type I (HTLV-1) [1]. ATL was first discovered in Japan, in 1977 [1,2]. HTLV-1 infections are endemic in some countries and regions, including Japan, Latin America, southwestern Africa, and some areas of Australia [3]. In Japan, there are more than one million carriers of HTLV-1 [4]; 4000 new cases of HTLV-1 infection per year [5]; approximately 2600 cases of newly diagnosed ATL within a 2 y period [6]; and 1000–1500 deaths from ATL annually [7]. ATL is usually categorized into four subtypes based on clinical findings: acute, lymphoma, chronic, and smoldering [8,9]. This classification is useful for making decisions concerning treatment. According to the criteria used to diagnose ATL [8], the indolent subtypes (i.e., smoldering and favorable chronic) are usually managed with watchful waiting until acute crisis [10], and the aggressive subtypes (i.e., acute, lymphoma, and unfavorable chronic) are managed using a variety of intensive chemotherapies followed by allogeneic hematopoietic stem cell transplantation (allo-HSCT) depending on the age [11,12]. More recently, anti-CC chemokine receptor 4 (CCR4) monoclonal antibody (mAb) (mogamulizumab) [13,14] and lenalidomide [15] have also been used for treating relapsed or refractory ATL. However, ATL generally exhibits a very poor prognosis. In fact, the recent 4-year survival rates (i.e., the median survival time, days) for acute-, lymphoma-, unfavorable chronic-, favorable chronic-, and smoldering-subtype ATL in Japan were 16.8% (252), 19.6% (305), 26.6% (572), 62.1% (1937), and 59.8% (1851), respectively [16]. Moreover, the 3-year overall survival rate for acute- or lymphoma-subtype ATL was reported to be 33% despite undergoing allo-HSCT, which is considered to be the only curative treatment, but which frequently causes severe adverse events [17]. It is, therefore, imperative to develop novel modalities for the treatment of ATL.

In Japan, the median age at diagnosis of ATL is 68 years old (interquartile range: 60–75 years old) [18]. In general, ATL onset requires a long latency period of approximately 50–60 years, which indicates the involvement of multistep mechanisms for leukemogenesis in HTLV-1-infected cells. The HTLV-1 proteins Tax and HBZ are involved in the alteration of immune traits in HTLV-1-infected cells, escape from host immune surveillance systems, and accumulation of genetic mutations [19,20,21,22,23]. It was reported that the frequencies of invariant natural killer T (iNKT) cells, NK cells, and dendritic cells in the peripheral blood of ATL patients were significantly decreased [24] and that NK cell activity was markedly low in ATL patients [25]. Recently, the adoptive transfer of autologous NK cells was reported to be effective and safe [26,27,28,29,30]. Immunotherapy is, therefore, considered to be a novel therapeutic and prophylactic strategy against HTLV-1 infections.

The adoptive transfer of T cells and NK cells has attracted much attention as a new strategy for cancer immunotherapy. Chimeric antigen receptor (CAR) T-cell therapy is a revolutionary new pillar in the treatment of B-cell lymphoma and multiple myeloma [31,32]. There are, however, many problems related to CAR T-cell therapy, such as life-threatening CAR T-cell-associated toxicities, limited efficacy against solid tumors, inhibition and resistance in B-cell malignancies, antigen escape, limited persistence, poor trafficking and tumor infiltration, and the immunosuppressive microenvironment [33]. With CAR NK cells, it is generally difficult to expand a large number of highly active NK cells for infusion therapies [26,29].

The aims of this study were to develop an efficient method for expanding autologous innate immune effector cells and to confirm whether the expanded cells exhibit potent cytotoxic activities against HTLV-1-infected cells in vitro. We focused on γδ T cells and NK cells as innate immune effector cells and examined the expansion rate and cytotoxicity against HTLV-1-infected cells.

γδ T cells are involved in an immune surveillance system against cancer, including hematological diseases and solid tumors [34,35,36,37,38,39,40,41,42,43,44,45]. γδ T cells occupy 3–5% of peripheral blood T cells, 50–75% of which express Vγ9 (also termed Vγ2)- and Vδ2-bearing T-cell receptors (TCRs) in healthy adults [46]. We hereafter use the term “γδ T cells” for Vγ9–Vδ2-bearing γδ T cells. Most γδ T cells express neither CD4 nor CD8 and do not require conventional antigen processing and presentation via major histocompatibility complex (MHC) molecules for antigen recognition. In addition, the majority of γδ T cells fail to recognize conventional peptide antigens. Instead, they recognize small phosphorylated metabolites, such as isopentenyl diphosphate (IPP) and dimethylallyl diphosphate (DMAPP), from the mevalonate pathway, as self-antigens and (*E*)-4-hydroxy-3-methylbut-2-enyl diphosphate (HMBPP) from the 2-*C*-methyl-d-erythritol 4-phosphate/1-deoxy-d-xylulose 5-phosphate (MEP/DOXP) pathway as a foreign antigen in a butyrophilin (BTN) 3A1/2A1-dependent manner [47,48]. Since the multifaceted properties of tumor cells depend on the spaciotemporal expressions of small G-proteins, such as Ras, Rap, and Rho, whose functions are inexorably linked to farnesyl and/or geranylgeranyl-groups derived from metabolites in the mevalonate pathway, tumor cells might contain an elevated level of IPP/DMAPP, which can be recognized by γδ T cells [49].

γδ T cells derived from the peripheral blood of young, healthy donors (hereafter referred to as HDs) can be efficiently expanded up to 95–99% for 10–11 days with tetrakis-pivaloyloxymethyl 2-(thiazole-2-ylamino) ethylidene-1,1-bisphosphonate (PTA), a nitrogen-containing bisphosphonate prodrug, and interleukin (IL)-2 [50]. Although IL-2 can expand NK cells, the extent is not sufficient for practical use in clinical settings. The IL-18 receptor is expressed on innate immune cells, such as γδ T cells and NK cells, and IL-18 could augment the proliferation of γδ T cells and promote the expansion of NK cells in the presence of IL-2, since IL-18 induces the expression of CD25, an IL-2 receptor α chain [51,52,53].

It is worth noting that a humanized anti-CCR4 mAb has been approved for the treatment of ATL, in which the mAb acts on CCR4-expressing HTLV-1-infected CD4^+^ T cells [54,55,56]. The therapeutic effect of anti-CCR4 mAb is considered to be partially dependent on antibody-dependent cellular cytotoxicity (ADCC) through FcγR IIIa (CD16) expressed on effector cells, such as γδ T cells and NK cells [51,55,57,58,59]. It is, therefore, intriguing to examine whether anti-CCR4 mAb enhances the cytotoxicity of γδ T cells and NK cells against HTLV-1-infected cells in vitro.

Since γδ T cells and NK cells are not restricted by MHC in the recognition of malignant cells, the allogeneic transfer of these innate immune cells is currently under investigation. It is, however, evident that autologous cell therapies are safer than allogeneic cell therapies, since residual αβ T cells might cause graft-versus-host diseases.

In this study, we examined the immunological properties of γδ T cells and NK cells derived from HDs, elderly non-ATL patients and ATL patients in an attempt to explore the possibility of the adoptive transfer of γδ T cells and NK cells in the treatment of ATL.

## 2. Materials and Methods

### 2.1. Derivation of γδ T Cells and NK Cells

γδ T cells were expanded from peripheral blood mononuclear cells (PBMCs) in Yssel’s medium supplemented with 10% heat-inactivated human AB serum [60], as described in Appendix A. NK cells were prepared from CD3-depleted PBMC, as described in Appendix A.

### 2.2. Flow Cytometric Analysis

Cells were stained with fluorescent dye-conjugated Abs, as described in Appendix A, and analyzed using a FACS Lyric flow cytometer (Becton Dickenson, Franklin, Lakes, NJ, USA). The cell population was visualized with FlowJo ver. 10.8.1 (FlowJo LLC, Ashland, OR, USA).

### 2.3. Patient Characterization and Outcome

This study was conducted in accordance with the Declaration of Helsinki and was approved by the Institutional Review Board of Nagasaki University Hospital. Obligatory written informed consent was obtained from each participant in accordance with the comprehensive prior consent given to the Departments of Hematology and Dermatology (Approval No. 13022512 and UMIN000042835).

In this study, 16 HDs with no known medical history (12 males and 4 females) were first enrolled. The median age at the time of blood sampling was 34 years (range: 27–58 years). γδ T cells and/or NK cells were expanded from the peripheral blood samples of 10 HDs, as shown in Appendix A. Then, 55 ATL patients were recruited in this study. Patients’ characteristics are summarized in Appendix A and the Shimoyama classification at the first diagnosis and the outcome at blood sampling are summarized in Appendix A. ATL patients were allocated to flow cytometric analysis, PTA/IL-2-induced expansion of γδ T cells, PTA/IL-2/IL-18-induced expansion of γδ T cells, and IL-2/IL-18-induced expansion of NK cells, which is summarized in Appendix A. In addition, elderly non-ATL patients (8 males and 2 females) were enrolled, as shown in Appendix A, since aging is reported to result in the remodeling of T-cell immunity and to be associated with poor clinical outcomes in age-related diseases [61]. They visited the Department of Dermatology, Nagasaki University Hospital between December 2021 and July 2023 and had no history of malignancy, HTLV-1 infections, and use of immunosuppressants or prednisolones. The median age at the time of blood sample collection was 71.5 years old (range: 66–92 years old), which is comparable to that of the ATL patients.

### 2.4. Cytotoxicity Assay Using Time-Resolved Fluorescence Spectroscopy

The cytotoxic activity of γδ T cells and NK cells against HTLV-1-infected cells was determined using a nonradioactive cellular cytotoxicity assay kit (Techno Suzuta Co., Ltd., Heiwa-machi, Nagasaki, Japan). The KK1 human ATL cell line was established in the Department of Hematology, Nagasaki University, and the HuT102 human ATL cell line was from American Type Culture Collection (ATCC, Manassas, VA, USA). The cell lines were maintained in complete RPMI1640 medium at 37 °C with 5% CO_2_ overnight. As for the ATL cell lines, 100 U/mL of IL-2 was added to the medium every other day. KK1 and HuT102 were pretreated with anti-CCR4 mAb (Kyowa Kirin Co., Ltd., Chiyoda-ku, Tokyo, Japan) (4 mg/mL stock solution in PBS) at final concentrations of 0.5 µg/mL and 10 µg/mL, respectively. The tumor cell suspensions (1 × 10^6^ cells in 1 mL of RPMI1640 medium) were then pulsed with 25 µM bis (butyryloxymethyl) 4′-(hydroxymethyl)-2,2′:6′,2″-terpyridine-6,6′-dicarboxylate (BM-HT, Techno Suzuta Co., Ltd.) at 37 °C for 15 min. When BM-HT was internalized in the tumor cells, the compound was hydrolyzed by intracellular esterases to yield 4′-(hydroxymethyl)-2,2′:6′,2″-terpyridine-6,6″-dicarboxylate (HT). Afterward, the cells were washed three times with 5 mL of complete RPMI1640 via centrifugation at 1700 rpm at 4 °C for 5 min. The tumor cell suspensions (5 × 10^3^ cells/100 µL) were dispensed into a 96-well round-bottom plate, to which was added 100 µL of a serial dilution of γδ T cells and/or NK cells. The plate was briefly centrifuged at 500 rpm at room temperature for 2 min and then incubated at 37 °C with 5% CO_2_ for 60 min. Detergent (Techno Suzuta Co., Ltd.) was added to each well at a final concentration of 5 × 10^−5^ M for maximum release, and the cell suspensions and the plate were incubated at 37 °C with 5% CO_2_ for 20 more min. After the cell suspensions were mixed well, the plate was centrifuged at 1700 rpm at 4 °C for 2 min. The supernatant, 25 µL each, was transferred to a new 96-well round-bottom plate containing 250 µL of europium (Eu^3+^) solution in 0.3 M sodium acetate buffer at pH 4 (Techno Suzuta Co., Ltd.), from which 200 µL each was transferred to Thermo Scientific 96-well plates. The time-resolved fluorescence (TRF) was measured using a NIVO multiplate reader (Revvity, Yokohama, Kanagawa, Japan). All measurements were performed in triplicate. The specific lysis (%) was calculated as 100 × (experimental release (counts) − spontaneous release (counts))/(maximum release (counts) − spontaneous release (counts)).

### 2.5. Statistical Analysis

Continuous data are presented as the median value, range, and interquartile range (IQR), and they were compared using Wilcoxon rank-sum tests with GraphPad Prism (version 10.0.2 for Windows, GraphPad Software, La Jolla, CA, USA). Categorical data were compared using Fisher’s exact tests. A *p*-value of less than 0.05 was considered to be statistically significant.

## 3. Results

### 3.1. Expansion of γδ T Cells and NK Cells Derived from HDs

We first expanded γδ T cells from PBMC of 10 HDs using PTA/IL-2, as shown in Appendix A. Four representative results of the flow cytometric analyses before and after expansion are shown in Appendix A. It is of note that a large number of highly purified γδ T cells were obtained by using a PTA/IL-2 expansion system [50], when the initial proportion of γδ T cells in the CD3^+^ lymphocyte fractions was well above 1%. The stimulated cells started to form clusters 3 to 5 days following the stimulation (Appendix A). NK cell-related cell surface markers [62,63,64,65,66,67], such as natural killer group 2 member D (NKG2D), DNAX accessory molecule-1 (DNAM-1), and CD16 (FcγRIIIA), were expressed on γδ T cells on day 11 (Appendix A). By contrast, the expression of FasL (CD95L) and TRAIL (human TNF-related apoptosis-inducing ligand) was marginal. PD-1 was expressed on γδ T cells to different degrees depending on individuals, which was consistent with previous reports [68,69].

We next expanded NK cells using IL-2 and IL-18 from 10 HDs. As shown in Appendix A, highly purified NK cells were obtained on day 10. The stimulated cells started to form clusters 4 to 5 days after stimulation with IL-2/IL-18 (Appendix A). The NK cells expanded for 10 days expressed high levels of NKG2D, DNAM-1, and CD16, as shown in Appendix A. In addition, more than half of the IL-2/IL-18-expanded NK cells expressed high levels of HLA-DQ [70] and CD86 [71].

### 3.2. Cytotoxicity Exhibited by γδ T Cells and NK Cells Derived from HDs against ATL Cell Lines In Vitro

We next examined the cytotoxic activity exhibited by PTA/IL-2-expanded γδ T cells against HTLV-1-infected cell lines with a time-resolved fluorescence (TRF)-based assay system using a terpyridine derivative and europium [72]. We examined the effect of anti-CCR4 mAb on the γδ T-cell-mediated cytotoxicity against KK1 and HuT102 HTLV-1-infected cell lines. As shown in Figure 1 (left panels), the γδ T cells exhibited potent cytotoxic activity against KK1 and HuT102 cells in an E/T ratio-dependent manner, with the specific lysis reaching approximately 20% in 60 min at an E/T ratio of 1:200. When 10 µg/mL of anti-CCR4 mAb was added to the assay system, 40 to 80% of either KK1 or HuT102 cells were killed by γδ T cells at an E/T ratio of 1:100, suggesting that the γδ T cells exhibited a potent ADCC against HTLV-1-infected cells in the presence of anti-CCR4 mAb (Figure 1, middle and right panels).

Following this, we examined the direct cellular cytotoxicity of the NK cells against HTLV-1-infected cells. As shown in Figure 2 (left panels), the specific lysis of NK cells against KK1 or HuT102 reached 20% to 70% in 60 min at an E/T ratio of 1:100. It is worth noting that the NK cells exhibited a more potent cellular cytotoxicity against HTLV-1-infected cells than the γδ T cells. When 10 µg/mL of anti-CCR4 mAb was added to the assay system, more than 40–80% of KK1 and HuT102 cells were killed by NK cells at an E/T ration of 1:100. Taken together, innate immune cells, including γδ T cells and NK cells derived from HDs, exhibited both a direct cellular cytotoxicity and ADCC against HTLV-1-infected cells.

### 3.3. Effect of IL-18 on the Expansion of γδ T Cells Derived from ATL Patients

We then examined the immunological properties of γδ T cells derived from ATL patients. The frequency of Vδ2^+^ T cells in CD3^+^ T cells was significantly low in the peripheral blood of 25 ATL patients compared to that of HDs, as shown in Figure 3A. When PBMCs derived from ATL patients were stimulated/expanded with PTA/IL-2 for 11 days, the γδ T cells proliferated well and the expansion rate was comparable to that of HDs, as shown in Figure 3B, which indicates that the PTA/IL-2-mediated expansion of the γδ T cells was not affected by age and HTLV-1 infection status in terms of the expansion rate.

In the studies on γδ T cells derived from HDs, it was difficult to obtain a large number of highly purified γδ T cells when the initial proportion of γδ T cells in the CD3^+^ lymphocyte fractions was too low, especially when the proportion was less than 1%. In the peripheral blood of ATL patients, in fact, the initial frequency of γδ T cells was mostly less than 1%. We therefore sought to develop a strategy to expand γδ T cells more efficiently even in the case of ATL patients whose proportion of peripheral blood γδ T cells was markedly low.

It was previously demonstrated that the IL-18 receptor is expressed on immune effector cells, such as NK cells, γδ T cells, and CD8^+^ killer T cells, and IL-18 could efficiently promote the expansion of γδ T cells [52] and NK cells [53] with potent cytotoxicity. We therefore expanded PBMCs derived from the same 25 ATL patients with PTA/IL-2/IL-18 for 11 days (under the same conditions, except for the addition of IL-18) and examined the proportion of γδ T cells in CD3^+^ T cells. As shown in Figure 3C, PTA/IL-2/IL-18 successfully amplified γδ T cells derived from 25 ATL patients, and the purity of the γδ T cells was higher than that for those expanded with PTA/IL-2. In addition, the expansion rate of the γδ T cells stimulated with PTA/IL-2/IL-18 was also greater than that for PTA/IL-2. We therefore enrolled 30 additional ATL patients and analyzed the PTA/IL-2/IL-18-mediated expansion of γδ T cells derived from a total of 55 ATL patients.

As shown in Appendix A, the initial proportion of γδ T cells was markedly low, compared to that of HDs. It is worthy of note that some CD4^+^ T cells expressed a slightly low level of CD3. It is most likely that the CD3^dim^CD4^+^ T cell population corresponds to ATL cells. A microscopic analysis revealed that the cells started to form clusters 3 to 6 days after stimulation depending on individuals (Appendix A). After expansion with PTA/IL-2/IL-18 for 11 days, the proportion of γδ T cells in lymphocyte fractions failed to reach to the levels for HDs, whereas the expansion rate was equivalent to that for HDs. It is intriguing that CD3^dim^CD4^+^ T cells mostly disappeared from the cell culture, suggesting that they were killed by γδ T cells. In fact, essentially all the expanded γδ T cells expressed NKG2D and DNAM-1, as shown in Appendix A. They expressed CD16 and PD-1 to different degrees depending on individuals. In addition, it is noteworthy that CD3^−^CD56^+^ cells were increased when the proportion of γδ T cells was low on day 11. It is most likely that this CD3^−^CD56^+^ cell population is NK cells, indicating that the ATL cells in the cell culture are also killed by NK cells.

Based on the above findings, it is essential to take the initial proportion of γδ T cells and the expansion of NK cells into account when we further explore the possibility of infusion therapy for ATL. As shown in Appendix A, CD3^−^CD56^+^ cells (corresponding to NK cells) were increased when γδ T cells failed to occupy the majority of lymphocytes after stimulation/expansion with PTA/IL-2/IL-18 for 11 days. Even in such cases, CD3^dim^CD4^+^ T cells (corresponding to ATL cells) disappeared after incubation for 11 days, strongly suggesting that PTA/IL-2/IL-18-expanded γδ T cells and NK cells could exert potent anti-ATL activity.

Hence, the ATL patients were divided into two groups: one that exhibited an initial frequency of γδ T cells in CD3^+^ T cells of less than 0.1% and one that was 0.1% or greater. In the group with a γδ T-cell frequency of less than 0.1%, the purity of the γδ T cells after expansion with PTA/IL-2/IL-18 for 11 days was significantly lower than that for the other group, as shown in Figure 4A. It is worth noting that the group with a lower proportion of γδ T cells tended to have a worse disease status, as determined using clinical indicators such as sIL-2R (U/mL), LDH (IU/L), BUN (mg/dL), WBCs (×10^9^/L), and Ab-Ly (%) (Figure 4B). Using Fisher’s exact tests, the low-frequency group had significantly more aggressive diagnoses (i.e., acute, lymphoma, and unfavorable chronic subtypes) based on the Shimoyama classification at the time of blood sampling (10 out of the 16 patients in this group, *p* = 0.0324), demonstrating that ATL patients with an aggressive-type diagnosis according to the Shimoyama classification tended to exhibit poor expansion of γδ T cells with PTA/IL-2/IL-18. In most cases, NK cells were expanded instead of γδ T cells after incubation with PTA/IL-2/IL-18. In three ATL patients who did not respond to PTA/IL-2/IL-18 at all, Ab-Ly occupied more than 90% of WBCs before expansion, and more than 90% of the cells remained ATL cells after expansion. However, such a poor expansion of γδ T cells and NK cells was observed in only a small number of the ATL patients, indicating that the development of infusion therapy using autologous γδ T cells and NK cells is feasible with most of the ATL patients.

### 3.4. IL-2/IL-18-Mediated Expansion of NK Cells Derived from ATL Patients

Since NK cells derived from HDs exhibited a potent cellular cytotoxicity against ATL cells and NK cells were expanded in the culture of ATL patient-derived PBMCs with a low proportion of γδ T cells in the presence of PTA/IL-2/IL-18, we set out to examine the effector functions of IL-2/IL-18-stimulated/expanded NK cells derived from 30 ATL patients. However, two particular cases with high Ab-Ly counts and extremely low CD3^−^ T lymphocyte counts were excluded from analysis.

As shown in Appendix A, the proportion of NK cells derived from ATL patients after expansion with IL-2/IL-18 for 10 days was comparable to that from HDs, whereas the expansion rate was significantly lower than that of HDs. The expression of NKG2D, DNAM-1, CD16, HLA-DQ, and CD86 in NK cells derived from ATL patients after expansion was equivalent to that of HDs.

### 3.5. Effect of Aging on the Phenotype and Immunological Properties of γδ T Cells and NK Cells

Since most ATL patients are elderly, it is essential to examine the effect of aging on the phenotype and immunological properties of γδ T cells and NK cells to distinguish the effect of the HTLV-1 infection status and age. We obtained peripheral blood samples from 10 elderly non-ATL patients, whose median age was comparable to that of ATL patients. After PBMCs were stimulated/expanded with PTA/IL-2/IL18 (see Supplementary Note added to Result Section 3.5), we compared cell surface markers on γδ T cells before and after expansion with PTA/IL-2/IL-18 between ATL patients and elderly non-ATL patients (Figure 5). The proportions of Vδ2^+^ T cells in CD3^+^ T cells in the peripheral blood of ATL patients were clearly low, possibly due to both aging and HTLV-1 infection status, while the proportion of Vδ1^+^ cells in CD3^+^ T cells remained unchanged, regardless of age and HTLV-1 infection status. However, with the exception of a few cases with extremely low levels of Vδ2^+^ T cells in CD3^+^ lymphocyte fractions, the expansion rates of Vδ2^+^ T cells from elderly, non-ATL patients were comparable to that of HDs and ATL patients (see also Figure 3B). No significant differences in the expression levels of CD16, NKG2D, DNAM-1, FasL, TRAIL, and PD-1 were observed between ATL patients and elderly non-ATL-patients.

We next stimulated CD3^−^ PBMC fractions derived from elderly non-ATL patients with IL-2/IL-18 for 10 days (see Supplementary Note added to Result Section 3.5), and compared cell surface markers before and after expansion among ATL patients, elderly non-ATL patients, and HDs (Figure 6). The proportions of NK cells in peripheral blood remained unchanged regardless of age and/or HTLV-1 infection status. After expansion with IL-2/IL-18, highly purified NK cells were obtained from CD3^−^ PBMC fractions of ATL patients, which was not influenced by age and/or HTLV-1 infection status. However, the expansion rate of NK cells in ATL patients tended to be low, compared to that in HDs, which might be due to aging and HTLV-1 infection status. No significant differences in the expression of CD16, NKG2D, DNAM-1, and CD86 were found between ATL patients and elderly patients.

### 3.6. Cytotoxic Activity Exhibited by γδ T Cells and NK Cells Derived from ATL Patients against ATL Cell Lines In Vitro

We next examined the cytotoxic activity of PTA/IL-2/IL-18-stimulated/expanded PBMCs containing γδ T cells and NK cells against KK1 and HuT102 HTLV-1-infected cells. After expansion of PBMCs derived from ATL patients with PTA/IL-2/IL-18, cell surface expressions of CD3, CD56, CD16, and Vδ2 were examined. As shown in Appendix A, the proportions of γδ T cells, NK cells, and CD16-positive cells varied to different degrees among ATL patients.

When KK1 or HuT102 cells were challenged by PTA/IL-2/IL-18-stimulated/expanded PBMCs containing various proportions of γδ T cells and NK cells, the specific lysis (%) reached 30% to 80% in 60 min at an E/T ratio of 1:200 (Figure 7). By adding 0.5 or 10 µg/mL of anti-CCR4 mAb, the specific lysis increased up to 40–80%, even at an E/T ratio of 1:100. It was worthy of note that the effect of anti-CCR4 mAb on the cellular cytotoxicity was associated with the proportion of γδ T cells, which was consistent with the observation in Figs. 1 and 2. Although the cytotoxicity of NK cells is higher than that of γδ T cells, the addition of anti-CCR4 mAb bolsters the cytotoxic activity of γδ T cells more efficiently, resulting in similar levels of cytotoxicity exhibited by the PTA/IL-2/IL-18-expanded γδ T cell and NK cell mixtures against HTLV-1-infected cells, even if the proportions of γδ T cells and NK cells vary to different degrees.

We next examined the cytotoxic activity of IL-2/IL-18-expanded NK cells derived from ATL patients against HTLV-1-infected cells (Figure 8). Flow cytometric diagrams of representative expansion patterns are depicted in Appendix A. When KK1 and HuT102 were challenged by IL-2/IL-18-expanded NK cells derived from ATL patients, the specific lysis reached to 20% to 80% in 60 min at an E/T ratio of 1:100. By addition of 0.5 or 10 µg/mL of anti-CCR4 mAb, the specific lysis was increased to 60–90% at an E/T ratio of 1:100 as shown in Figure 8. On the basis of these results, NK cells derived from ATL patients exhibited a highly potent and stable cytotoxicity against HTLV-1-infected cells, as in the case of HDs. Taken together, it is most likely that the PTA/IL-2/IL-18-expanded γδ T cells and NK cells are ideal immune effector cells for adoptive cell therapy against ATL.

## 4. Discussion

Although infusion therapies for ATL using innate immune cells, such as NK cells [26] and CAR iNKT cells [73], have been developed extensively over the past few years, it is still a long way off from clinical use in terms of their efficacy and cost. In this study, we explored the possibility of the development of γδ T cells/NK cells-based adoptive transfer therapy for ATL.

γδ T cells belong to both innate immunity and adaptive immunity and are involved in the first line of defense against cancer including solid tumors [34,35,36,37,38,39,40,41,42,43,44,45] and lymphoid malignancies [74]. Previous reports [50] and the present study demonstrated that γδ T cells could be readily amplified in HDs and the expanded γδ T cells exhibited potent cytotoxicity against HTLV-1-infected cells. ATL patients are, however, generally elderly and infected with HTLV-1 viruses, and the immune system in ATL patients is mostly suppressed [24,25,26]. Thus, our primary question was how aging and the state of immunosuppression affect the expansion and immunological properties of γδ T cells.

It was noteworthy that the proportion of peripheral blood γδ T cells from ATL patients was extremely low, with the median proportion of γδ T cells being 0.29% (range: 0.0–7.41%). In fact, the initial frequency of γδ T cells in CD3^+^ lymphocyte fractions was less than 0.1% in 29.6% of ATL patients, whereas such a low frequency of γδ T cells was observed only in 10% of elderly non-ATL patients and 0% of HDs. On the basis of these findings, the low proportion of peripheral blood γδ T cells seemed to be attributable to aging and the state of immunosuppression. In fact, when the state of ATL disease was worse in terms of clinical indicators and aggressive diagnoses in Shimoyama classification, the proportion of γδ T cells in the peripheral blood tended to be lower. It was, however, worthy of note that the expansion rate of γδ T cells in ATL patients was comparable to that of HDs. Even if the expansion rate is high enough, the initial proportion directly affects the number of γδ T cells after expansion, leading to a difficulty in the preparation of a large number of γδ T cells ex vivo. It is, therefore, prerequisite to develop a strategy to increase the number of γδ T cells more efficiently.

It was previously demonstrated that IL-18 could efficiently augment the expansion of γδ T cells and NK cells with potent cytotoxicity [51,52,53]. When PBMCs were incubated with IL-18 in addition to PTA/IL-2, the expansion rate of the γδ T cells was significantly greater than that with PTA/IL-2. In addition, NK cells were also expanded in the culture due to the presence of IL-2/IL-18. Although NK cells increased in number with stimulation using IL-2/IL-18, the expansion rate was generally low compared to that of the γδ T cells, even in HDs. The present study demonstrated that NK cell proliferation, itself, tended to reduce with aging, whereas the expressions of NKG2D, DNAM-1, CD16, HLA-DQ, and CD86 remained unchanged.

Although γδ T cells failed to proliferate well even in the presence of PTA/IL-2/IL-18, in some cases, an increase in the number of NK cells was observed, instead, in the culture, suggesting that the NK cells were complementarily expanded when the initial proportion of γδ T cells was extremely low. Since NK cells also exhibit anti-ATL activity, we should consciously examine the expansion of NK cells in culture that is intended to expand γδ T cells for use in adoptive cell therapy.

We next examined the cytotoxicity of the expanded innate immune effector cells against HTLV-1-infected cells. When HTLV-1-infected cells were challenged by γδ T cells and NK cells, the direct cytotoxicity exhibited by NK cells was much higher than that by γδ T cells, suggesting that the NK cells were superior to the γδ T cells as effector cells in terms of cytotoxicity. It is, however, difficult to prepare a large number of NK cells for use in infusion therapy, compared to γδ T cells, indicating that γδ T cells are superior to NK cells when it comes to the preparation of effector cells. There are advantages and disadvantages to both cell types.

In studies on the effect of IL-18 on the anti-CD20 mAb-mediated regression of non-Hodgkin lymphoma, it was demonstrated that IL-18 synergized with the mAb in the lymphoma regression [75]. Since both NK cells and γδ T cells express CD16, it is intriguing to examine whether ADCC plays a certain role in the regression. As for ATL, HTLV-1-infected cells express a high level of CCR4, implicating that the inclusion of anti-CCR4 mAb might enhance the cytotoxic activity of NK cells and γδ T cells. When the cytotoxic activity of the mixture of γδ T cells and NK cells derived from ATL patients was measured, the immune effector cell mixtures exhibited high levels of cellular cytotoxicity against HTLV-1-infected cells to different degrees. When anti-CCR4 mAb was added to the system, the cytotoxicity was enhanced to different degrees, depending on the proportion of γδ T cells. When the proportion of γδ T cells was relatively high, the add-on effect of the mAb was more prominent, suggesting that ADCC mediated by γδ T cells was more efficient than that by NK cells, since the direct killing of HTLV-1-infected cells by NK cells was intrinsically high even in the absence of the ADCC pathway.

In this study, it was clearly demonstrated that HTLV-1-infected cells were susceptible to the cytotoxic activity of γδ T cells in vitro, which was further enhanced by the addition of anti-CCR4 mAb. In addition, NK cells exhibited potent cytotoxicity against HTLV-1-infected cells even in the absence of mAbs. In order to move on to clinical trials, the present in vitro findings should be confirmed in animal models using an immunocompromised mouse model. In this study, three ATL patients failed to respond to PTA/IL-2/IL-18 at all. In these patients, the proportion and the number of HTLV-1-infected cells were too high and the cell culture for the expansion of NK cells and γδ T cells could not be appropriately set up. It should be thus considered that the removal of HTLV-1-infected cells should be included in the protocol for the expansion of NK cells and γδ T cells.

Since the present study was only a preliminary investigation to evaluate the feasibility of adoptive transfer therapy harnessing γδ T cells and NK cells, an autologous system was considered, with the safety of the treatment being the most important issue. The adoptive transfer of allogeneic γδ T cells is more practically promising, because it is much easier to expand γδ T cells and NK cells derived from HDs. In fact, γδ T cells and NK cells derived from HDs could be adoptively transferred into elderly and immunocompromised ATL patients, because the γδ T cell- and NK cell-mediated cytotoxicity is not restricted by the MHC molecules. It is, however, essential to conduct clinical trials carefully and extensively for the development of such allogeneic innate immune effector cell-based infusion therapy for ATL. Taken together, autologous γδ T cells and NK cells could be utilized for the treatment of ATL. If the safety and efficacy of allogeneic γδ T cells and NK cells is proven in clinical trials, γδ T cells and NK cells derived from HDs could be used together with anti-CCR4 mAb for the treatment of ATL as an over-the-counter treatment in the near future.

## 5. Conclusions

In this study, we have shown the potential of a new therapeutic strategy for ATL. In the future, we would like to conduct in vivo studies in a clinical setting and create a system that enables allogeneic γδ T cell transplantation.

## Figures and Tables

**Figure 1 cells-13-00128-f001:**
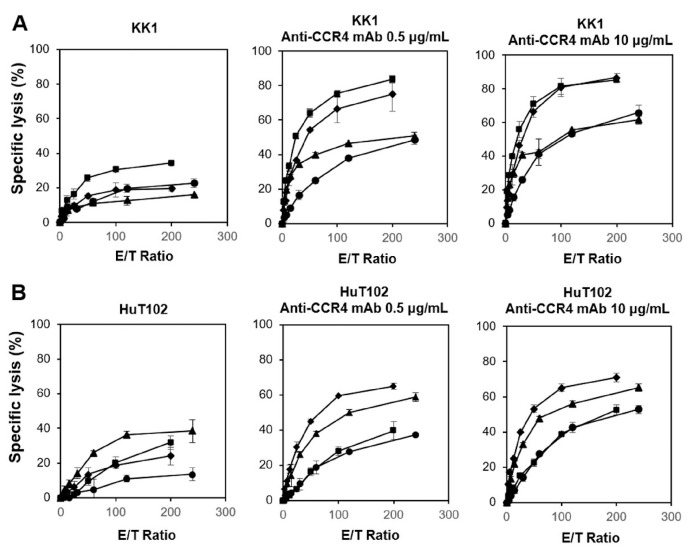
Cytotoxic activity exhibited by γδ T cells derived from HDs against HTLV-1-infected cells. Effect of anti-CCR4 mAb on the cytotoxic activity of γδ T cells against KK1 (**A**) and HuT102 (**B**). After HTLV-1-infected cell lines were pretreated with 0, 0.5, or 10 μg/mL of anti-CCR4 mAb for 15 min, the sensitized cells were challenged with PTA/IL-2-stimulated/expanded γδ T cells derived from four HDs at E/T ratios of 3.90625:1, 7.8125:1, 15.625:1, 31.25:1, 62.5:1, 125:1, and 250:1 or 3.125:1, 6.25:1, 12.5:1, 25:1, 50:1, 100:1, and 200:1 for 60 min, and the specific lysis was determined using a time-resolved fluorescence-based assay based on an europium–terpyridine derivative chelate. The various symbols depict the cytotoxic activity of γδ T cells from distinct HDs.

**Figure 2 cells-13-00128-f002:**
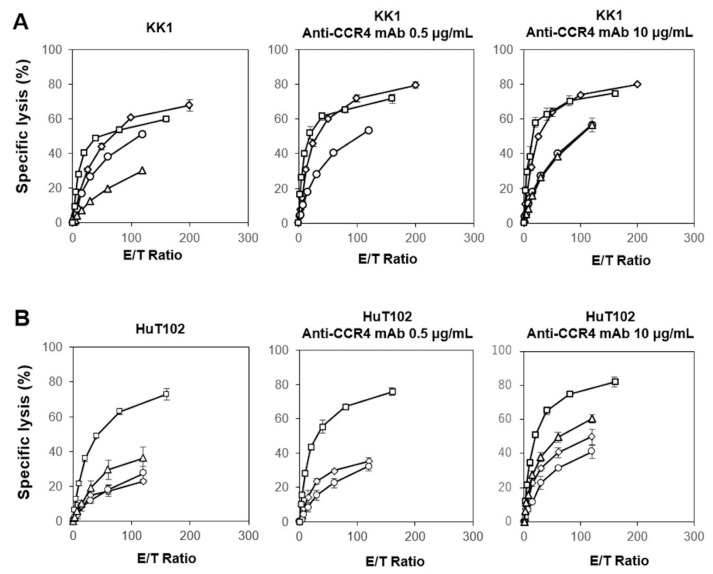
Cytotoxic activity exhibited by NK cells derived from HDs against HTLV-1-infected cells. Effect of anti-CCR4 mAb on the cytotoxic activity of NK cells against KK1 (**A**) and HuT102 (**B**). After HTLV-1-infected cell lines were pretreated with 0, 0.5, or 10 μg/mL of anti-CCR4 mAb for 15 min, the sensitized cells were challenged with PTA/IL-2-stimulated/expanded γδ T cells derived from four HDs at E/T ratios of 2.5:1, 5:1, 10:1, 20:1, 40:1, 80:1, and 160:1 or 1.875:1, 3.75:1, 7.5:1, 15:1, 30:1, 60:1, and 120:1 for 60 min, and the specific lysis was determined using a time-resolved fluorescence-based assay based on an europium–terpyridine derivative chelate. The various symbols depict the cytotoxic activity of γδ T cells from distinct HDs.

**Figure 3 cells-13-00128-f003:**
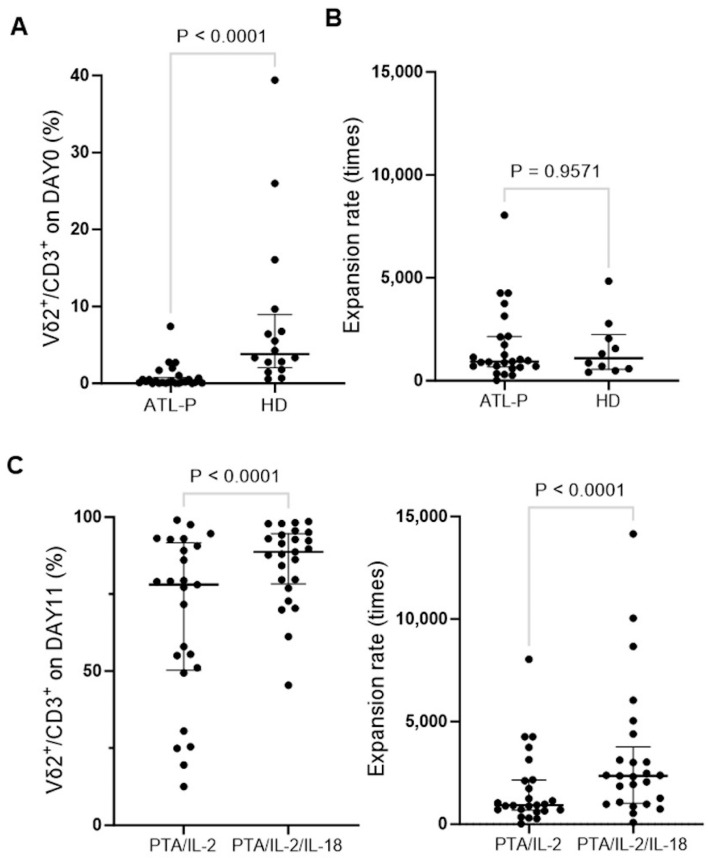
Comparison of the γδ T-cell expansions between ATL patients and HDs. (**A**) Proportion of γδ T cells in CD3^+^ T cells before expansion. PBMCs from 25 ATL patients and 10 HDs were purified from peripheral blood samples and stained with PE-conjugated anti-CD3 mAb and FITC-conjugated anti-Vδ2 mAb, which were analyzed using a FACS Lyric flow cytometer. (**B**) Expansion rate of γδ T cells in response to PTA/IL-2. After stimulation/expansion with PTA/IL-2 for 11 days, the cells were stained and analyzed as in (**A**) and counted under a microscope to calculate the number of γδ T cells. (**C**) Comparison of PTA/IL-2 and PTA/IL-2/IL-18 in the expansion of γδ T cells. PBMCs were stimulated/expanded with either PTA/IL-2 or PTA/IL-2/IL-18, and the effect of IL-18 was examined using flow cytometric analyses and the cell counting method.

**Figure 4 cells-13-00128-f004:**
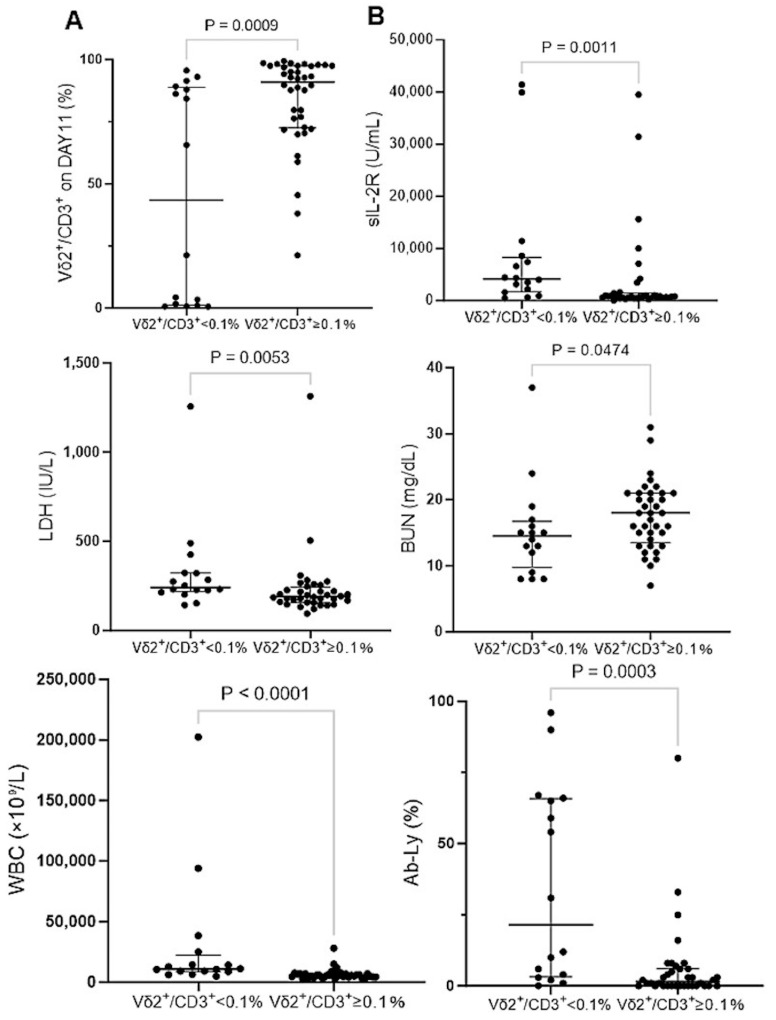
Effect of the initial proportion of Vδ2^+^ T cells in CD3 cells on the expansion of γδ T cells and comparison of blood test results between ATL patients with relatively high levels of initial Vδ2/CD3 ratio (greater than 0.1%) and those with low levels (less than 0.1%). (**A**) Proportion of Vδ2^+^ cells in CD3 cells after expansion of PBMCs with PTA/IL-2/IL-18. After stimulation/expansion of PBMCs derived from ATL patients with PTA/IL-2/IL-18, the cells were analyzed through a FACS Lyric flow cytometer. (**B**) Comparison of blood test results stratified by the initial proportion of Vδ2^+^ T cells in CD3 cells. Laboratory test results were compared based on the initial proportion of Vδ2^+^ T cells in CD3^+^ T cells.

**Figure 5 cells-13-00128-f005:**
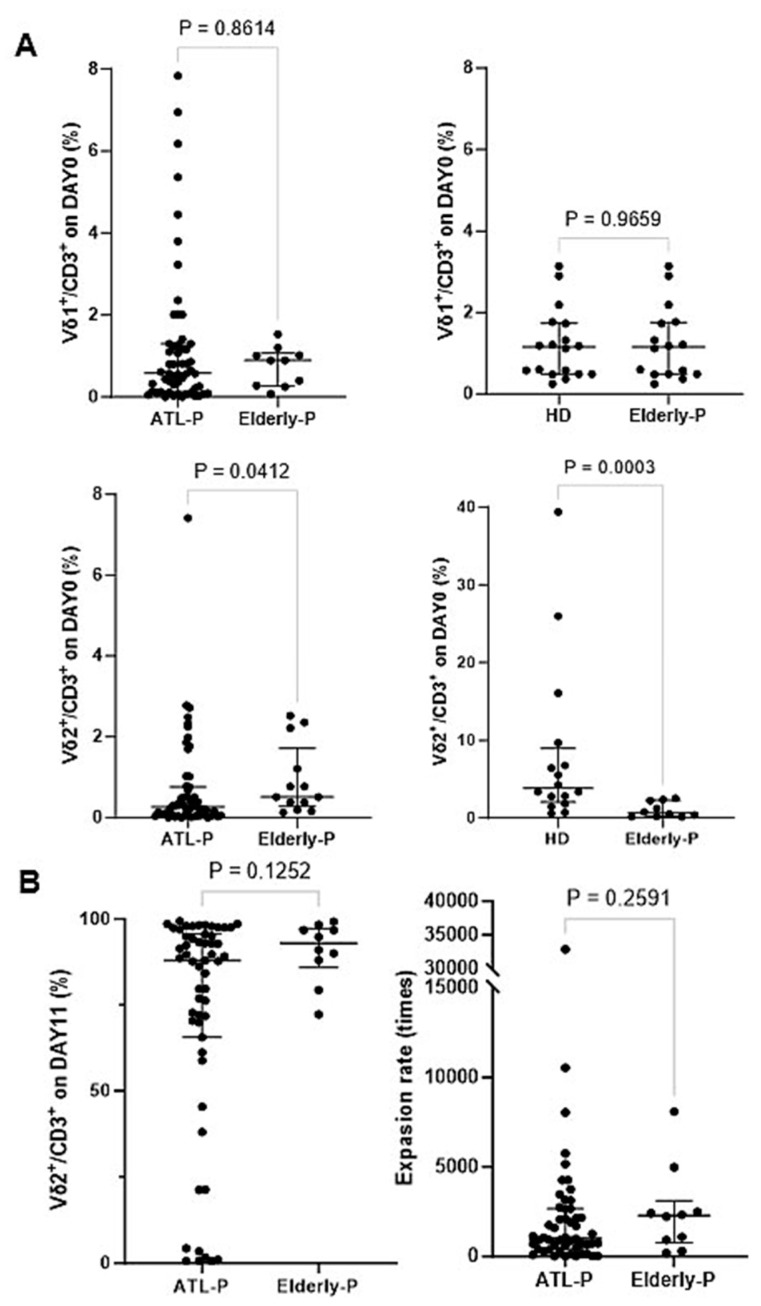
Comparison of PTA/IL-2/IL-18-mediated expansion of γδ T cells among ATL patients, elderly non-ATL patients, and HDs. (**A**) Proportions of Vδ1^+^ T cells and Vδ2^+^ T cells in CD3^+^ T cells before expansion; these initial proportions were compared among the three groups. (**B**) Proportion of Vδ2^+^ T cells in CD3^+^ T cells after expansion with PTA/IL-2/IL-18, and the expansion rates of the Vδ2^+^ T cells were compared between ATL patients and elderly non-ATL patients.

**Figure 6 cells-13-00128-f006:**
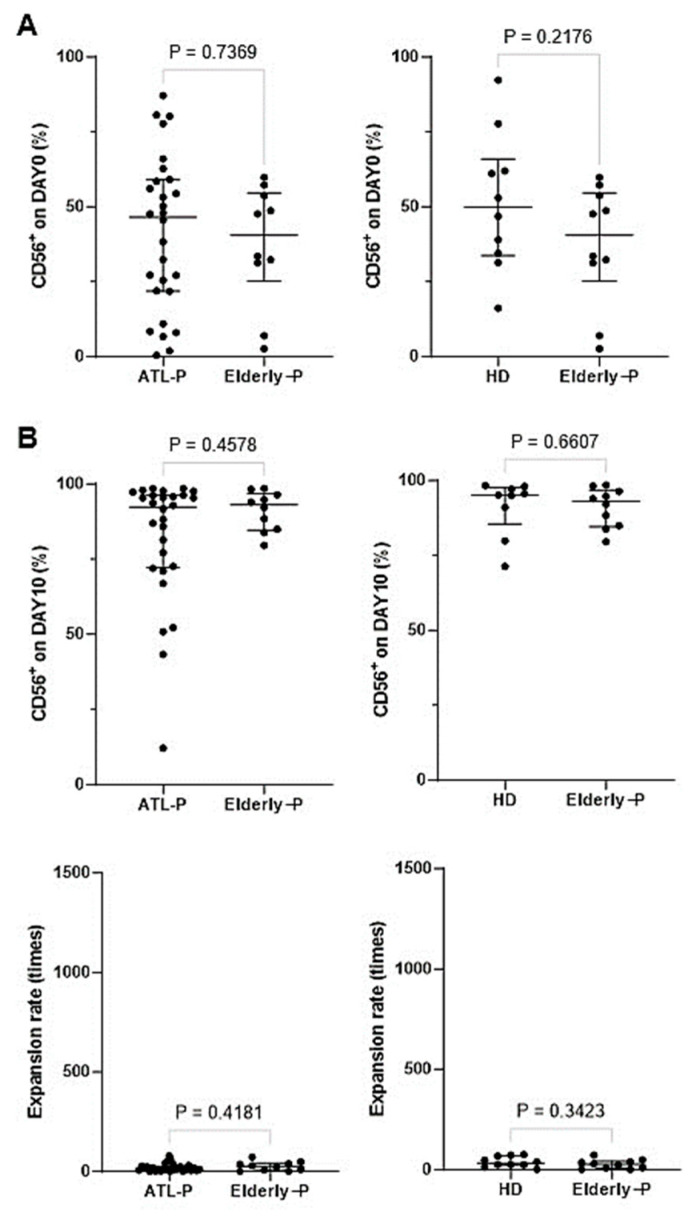
Comparison of NK cells among ATL patients, elderly non-ATL patients, and HDs. (**A**) Initial proportion of NK cells before expansion. The initial proportions of CD56^+^ cells in PBMCs were compared among the three groups. (**B**) Proportion of NK cells after expansion with IL-2/IL-18. After the expansion, the proportions of CD56^+^ cells were compared among the three groups.

**Figure 7 cells-13-00128-f007:**
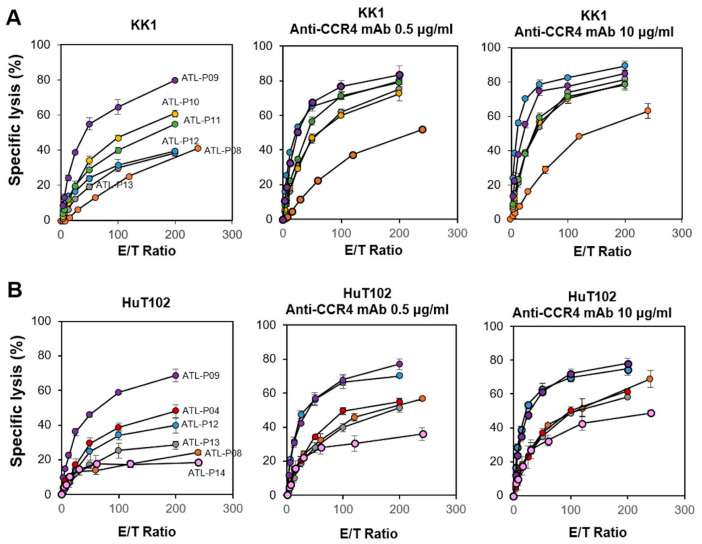
Cytotoxic activity exhibited by γδ T cells and NK cells derived from ATL patients against HTLV-1-infected cells. Effect of anti-CCR4 mAb on the cytotoxic activity of γδ T cells against KK1 (**A**) and HuT102 (**B**). After HTLV-1-infected cell lines were pretreated with 0, 0.5, or 10 μg/mL of anti-CCR4 mAb for 15 min, the sensitized cells were challenged with PTA/IL-2/IL-18-stimulated/expanded γδ T cells and NK cells derived from patients ATL-P04 and ATL-P08-14 at E/T ratios of 3.75:1, 7.5:1, 15:1, 30:1, 60:1, 120:1, and 250:1 or 3.125:1, 6.25:1, 12.5:1, 25:1, 50:1, 100:1, and 200:1 for 60 min, and the specific lysis was determined through a time-resolved fluorescence-based assay based on an europium–terpyridine derivative chelate.

**Figure 8 cells-13-00128-f008:**
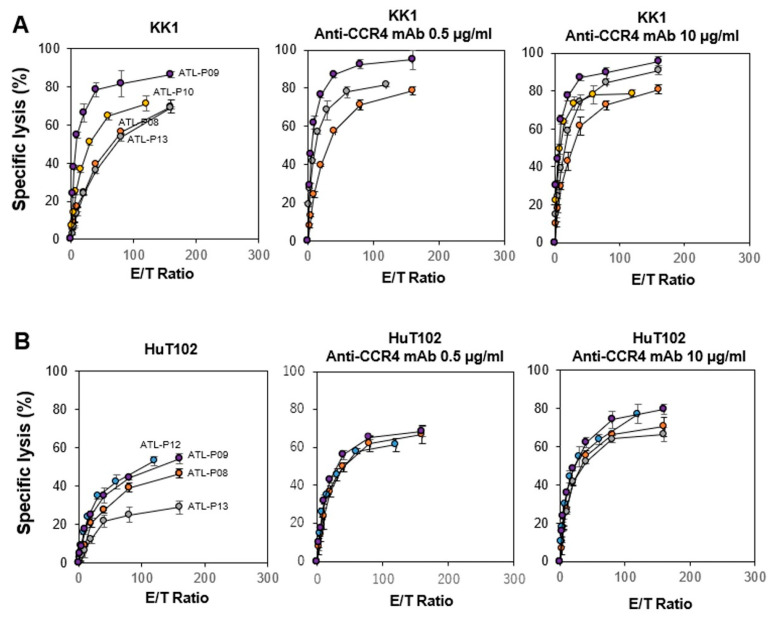
Effect of anti-CCR4 mAb on the cytotoxic activity of NK cells derived from ATL patients against KK1 (**A**) and HuT102 (**B**). After HTLV-1-infected cell lines were pretreated with 0, 0.5, or 10 μg/mL of anti-CCR4 mAb for 15 min, the sensitized cells were challenged with IL-2/IL-18-stimulated/expanded NK cells derived from patients ATL-P08-10 andATL-P12-13 at E/T ratios of 1.875:1, 3.75:1, 7.5:1, 15:1, 30:1, 60:1, and 120:1 or 2.5:1, 5:1, 10:1, 20:1, 40:1, 80:1, and 160:1 for 60 min, and the specific lysis was determined using a time-resolved fluorescence-based assay based on an europium–terpyridine derivative chelate.

## Data Availability

The raw data supporting the conclusions of this article will be made available by the authors, without undue reservation.

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
