# Peer review of "Development of Innate-Immune-Cell-Based Immunotherapy for Adult T-Cell Leukemia–Lymphoma"

_cells, 2024, doi:10.3390/cells13020128_

Round 1

Reviewer 1 Report

Comments and Suggestions for Authors

The authors present here a very complex and comprehensive work, where analyze subpopulations of T- and NK cells in the context of ATL. The paper has 22 pages of the text (without references), 9 Figures (including many subfigures) and other 9 supplementary Figures (again containing many subfigures). Despite huge amount of work, this paper is very difficult to read and contains also many inaccuracies and maybe mistakes.

My major remarks:

1) In the Abstract, I do not think (according to paper), that the authors really "set out to develop infusion therapy", I think more, that the authors tried to test in vitro therapetic potential of somehow expanded T and Nk cells.

2) In the Introduction, there is necessary to clearly write, what were aims of this study/project. (eg. ....). I should say, that firstly in was testing of gama/delta and NK cells in different subpopulations (healthy young/elderly and ATL patients), secondly testing effects of some drug (IL2, IL-18PTA and antiCCR4) on levels and activity of gama/delta and NK cells in the context of ATL in vitro (using cell-lines and cells from the real patients).

Why the authors choose these concrete drugs (IL-2, PTA, IL-18 and antiCCR4)? Please explain and eventually support by evidence.

2) In the Methods, there is nothing about proper study design (prospective, retrospective)?, how were 55 ATL patients selected?.  How long is follow up of patients? Also the numbers and brief description of "healthy donors" samples belongs into Methods.

3) Results are presented in a very bad way unfortunatelly, there is too much repeated information (again repeated in the Discussion) spread accross the paper. I recommend to re-structured the results, use shorter and more concise style with reduction of volume of text and also Figures (18 Figures with many subfigures is not instructive at all)

4) Moreover I cannot count well numbers of tested patients. The total number of ATL patients enrolled were 55, from them 31 untreated, but there 20 paients with chemo and other 7 with other therapy....that makes together 58 patients totally- please correct or explain better.

5)I have the similar problem with counting of 55 ATL patients analyzed for gama/delta levels (25 pts only???) and for NK cells (n=28???). Please explain if all patients were tested, or explain why was tested a part of them only. 

Also numbers of tested healthy young donors are not clear. For example 12+4 youngs were tested for gama/delta cells, but 10 youngs only for NK. Why?

How many (absolute counts) were ATL patients with low ( less than 0,1%) and high frequency of gama/delta cells?

In the Results, there are many sentences, which would fit into Introduction or Discussion, For example paragraph 3.3 (lines 391-403).

6) Discussion is very badly written. The obtained results should be discussed with similars data of published literature. In this paper, a lot a facts are repeated from Introduction or Results but without some evidences. Moreover, there are practically no references cited in this section (usually main part of references is related just to Discussion).

7) References - I think too much (eg. references 19-27 related to one sentence???), on the other hand I cannot find relevant text to references  37-40.

Minor comments:

Mistakes in writing exponents/indexes, many words are divided wihout being at the end of the line. Sometimes Figures do not correspond with relevant reference in the text, abbreviations are not clearly explained.  

Comments on the Quality of English Language

There are many mistakes in stylistics and ortography. 

Reviewer 2 Report

Comments and Suggestions for Authors

The article concerns an original approach to immune therapy of HTLV-caused T-cell leukemia (ATL) using adoptive transfer of γδ T cells and NK cells. The study was performed with blood samples from 55 ATL patients stimulated in vitro with bisphosphonate prodrug (PTA) and interleukin (IL)-2/IL-18), then testing for cytotoxicity with continuous ATL cell strains. The specific FC markers of γδ T cells and NK cells were applied in proper way. The novelty of work is determined by in vitro supplementation of an original biphosphonate prodrug compound.

The in vitro expansion was mainly observed in  γδ T cell populations. Of clinical and laboratory findings, the effect of aging on the phenotype of potentially effector γδ T cells and NK cells (line 556) is of special interest.

(Anti-CCR4 antibodies enhanced the cytotoxic activity of γδ T cells).

Some remarks and questions are as follows:

Introduction (line 103) “It has been reported” could be replaced by “…In this study we report that…”

Line 128: ‘the PTA stock solution was in DMSO’ -  one could note  final DMSO concentrations in cell cultures, since DMSO itself is known to exert sufficient cytotoxic and differentiogenic effects.

Line 162-163: ‘IL-2 and IL-18’: manufacturer (or purification procedure) of these cytokines should be noted

Line 193:… The Review Board … granted an exemption from the requirement for written informed consent… could be replaced by … granted a release from obligatory written informed consent …, since this sentence contradicts the statement in line 835

Line 196: Median and age ranges should be noted for the elderly participants of this study.

Line 207: …’relatively non-ATL patients’… what patients are meant?  Number of cases?

Line 265: please, decipher here ‘HD01–04’ abbreviation (healthy donors?)

Results: line 337: γδ T cells derived from a representative young healthy donor… In this case a short text description will be enough.  Therefore, Fig.1 may be skipped. Since quite compelling results are shown on Fig.2.

Line 337 ‘Clinical profiles of ATL patients’ may be placed to Materials and Methods, in order to present clinical pattern of the patients observed.

Discussion: this section lacks the data on probable mechanisms of cytotoxicity-inducing effects of PTA and its mixture with  IL-2 and IL-18. What were the reasons for choosing this combination of inducers?

A number of misprints are noted throughout the text: (line 109: ‘col-lected’, line 161 ‘sus-pension’; line 194 ‘Der-matology’, line 289: …worthy… of note, line 389: ‘expansion fold’ – could be ‘expansion rate’, etc.)

Sufficient copy editing is required to make some sentences understandable.

The article is quite interesting, contains good results, but requires major revision before publication in order to clarify some textual and technical ambiguities.

Reviewer 3 Report

Comments and Suggestions for Authors

The manuscript reports a potential therapeutic strategy by using γδ T cells and natural killer (NK) cells for adult T-cell leukemia-lymphoma (ATL). The data demonstrated in vitro expansion  of γδ T cells and NK cells from ATL patients or from heathy donors after stimulation with PTA/IL-2/IL-18. Anti-tumor activities of these two types of cells in ATL are shown by an in vitro cytotoxicity assay. Regarding translational perspectives, the significance of the in vitro analyses is very limited. The study could be strengthened by in vivo studies, such as using an immunocompromised mouse model. 

Comments on the Quality of English Language

Proofreading is necessary as many typos are across the manuscript especially in the Materials and Methods.

Round 2

Reviewer 1 Report

Comments and Suggestions for Authors

Development of innate immune cell-based immunotherapy for adult T-cell leukemia-lymphoma

Dear Editor,

I read the revised version of above-mentioned paper, and I am disappointed. The authors made only very cosmetic changes; some of my remarks were not explained or answered at all. In some cases, the authors repeated the criticized sentences only!

I cannot word by word correct and comment all the paper. I will put some examples only.

1)      The original article had 22 pages, it has now after reduction and correction 23 ages (without references)

2)      Reduction of Figures: from nine to eight …. and Supplementary Figures from nine to six! I think, that many of them can be omitted, because the same data are described in detail in the text.

3)      In the text, there is lasting confusion of methods, introduction and results. For example,

Methods: Lines 248-252 - the practically same text like in Introduction (lines 67-75)

Results (par 3.1): Lines 329-331 – this sentence would better belong into Discussion….

Results (par 3.1): lines   347-353 – better in the Introduction or partially in Methods

Results (par 3.1): lines 381-384 – better in the Intro

4)      Again, there is no explanation for discrepancy:

Methods (lines 222-225)“ We first prepared highly purified γδ T cells or NK cells derived from 16 young, healthy donors with no known medical history (12 males and 4 females). The median age at the time of blood sampling was 34 years (range: 27 – 58 years). γδ T cells and NK cells were expanded from the peripheral blood samples of 10 healthy donors

-OK, I can understand that for some (technical?) reasons, the expanded samples were 10 only.

But in the Results is written (lines 356-357)

„We next expanded the NK cells using IL-2 and IL-18, as previously reported. PBMC samples were obtained from 10 young, healthy volunteers, and CD3+ cells were depleted using anti-CD3 mAb-coated beads.“

-          According this sentence, it seems, that 10 samples were selected (how?) from initially 16 before expansion…..Authors, unfortunately, did not explain the reason or describe the algorithm better…

5)      I am very sory, but Table 1 (corrected) is going on to be confusing….

For example: one section of Table is called “Previous treatments at the time of blood sampling”, but its subsection is called “Undergoing anticancer drug treatment” – that is not very logical….., moreover I cannot understand what the authors mean by “post anticancer drug treatment” (? Patients with ended therapy in remission?) and “Breakdown of anticancer drug treatment”. Generally, from scientific point of view, it is very unhappy to mix patients from different phases of disease/therapy together - untreated with somehow treated with after therapy (in remission? or with active progressive disease…?). It also can influence experiment results.

6)      Discussion is better, but again, I would welcome to put the particular part of the results a long with similar data from literature, if they are not published yet, this fact is important to note too.  

Comments on the Quality of English Language

see above

Reviewer 2 Report

Comments and Suggestions for Authors

The revised version of article is characterized by more detailed presentation, especially, in Materials and Methods, taking into account the remarks on details of cell culture and clinical characteristics of the patients. The schedule and reasons of this approach become more understandable. In Discussion, the aims of in vitro research are more clearly formulated. In particular, the reasons and possible factor of combined in vitro combined effects of PTA+cytokine cocktails are discussed, thus meeting another major objection of reviewer. Copy editing is also performed making the reading process more smooth.

In sum, the main editorial issues of the paper seem to be resolved, thus recommending this interesting in vitro study for publication.

Comments on the Quality of English Language

The text is readable, only final copy reading is required

Reviewer 3 Report

Comments and Suggestions for Authors

In the revised manuscript, typos were corrected and some confusing sentences were reworded. 
